# Piezoresistive Behavior of a Conductive Polyurethane Based-Foam for Real-Time Structural Monitoring

**DOI:** 10.3390/s23115161

**Published:** 2023-05-29

**Authors:** Antoine Poirot, Nacera Bedrici, Jean-Christophe Walrick, Michel Arrigoni

**Affiliations:** 1ESTACA, ESTACA’Lab—Laval, 53000 Laval, France; nacera.bedrici@estaca.fr (N.B.); jean-christophe.walrick@estaca.fr (J.-C.W.); 2ENSTA-Bretagne, IRDL, UMR 6027 CNRS, 2 rue François Verny, 29806 Brest, France

**Keywords:** piezoresistivity, impact sensing, polyurethane foam, structural health monitoring (SHM), smart materials

## Abstract

Smart flexible materials with piezoresistive property are increasingly used in the field of sensors. When embedded in structures, they would allow for in situ structural health monitoring and damage assessment of impact loading, such as crash, bird strikes and ballistic impacts; however, this could not be achieved without a deep characterization of the relation between piezoresistivity and mechanical behavior. The aim of this paper is to study the potential use of the piezoresistivity effect of a conductive foam made of a flexible polyurethane matrix filled with activated carbon for integrated structural health monitoring (SHM) and low-energy impact detection. To do so, polyurethane foam filled with activated carbon, namely PUF-AC, is tested under quasi-static compressions and under a dynamic mechanical analyzer (DMA) with in situ measurements of its electrical resistance. A new relation is proposed for describing the evolution of the resistivity versus strain rate showing that a link exists between electrical sensitivity and viscoelasticity. In addition, a first demonstrative experiment of feasibility of an SHM application using piezoresistive foam embedded in a composite sandwich structure is realized by a low-energy impact (2 J) test.

## 1. Introduction

These last decades, composite structures have become essential in many industrial applications. As a key material for eco-design in the transport industry (aeronautics, space, defense), civil engineering or even energy production (wind turbine, pipelines), they are facing a variety of loadings (collision, vibration, flexion, etc.) during their lifetime. In this context, where safety standards are particularly drastic, numerous research works aim to improve structure performance in terms of weight, manufacturing, safety and durability. Sandwich structures made from composite materials are a representative example [1,2]. They are particularly used for their energy absorption properties combined with their light weight and high stiffness. Nevertheless, these structures are vulnerable when facing impacts, which could create an internal damage, such as delamination or kissing bonds, with serious consequences. Even though numerical models were created [3], the total prevention of risks remains challenging. Therefore, structures in general are regularly controlled during maintenance operations with various techniques. Among them is structural health monitoring (SHM). This technique consists of a real-time identification of eventual damages by making a health-state diagnostic of a structure by embedding numerous sensors (optical fibers, piezoelectric or piezoresistive sensors, etc.) onto or within it [4]. Depending on the sensors’ nature and structure type, SHM operations can be complex, time-consuming and expensive, such as, for example, the immobilization of a commercial aircraft for maintenance. For those reasons, the new tendency in the materials field is to create smart composite structures that allow for real-time monitoring. Therefore, composite materials embedding piezoelectric, capacitive, optical fibers or piezoresistive sensors in their conception have been tested [4,5,6,7,8]. The main problem of these new structures is the sensors’ integration, which possibly causes structure degradation. Moreover, even if traditional sensors have a good performance, they are too complex to produce because they are mostly designed with metals or inorganic semiconductors, which also reduce their flexibility [7,9]. 

Therefore, over the last decade, new soft sensors made from smart materials have been highly studied. Smart-polymer-based composites, such as smart textiles, piezoelectric elastomers or piezoresistive foams, are solutions of interest for strain/pressure sensing [9,10,11]; however, to the authors’ knowledge, no research works have reported on the utilization of piezoresistive foams as sensors for SHM application.

Thanks to their microcellular morphology and low density, the foams are ultra-light and highly compressible making them valuable as low intrusive materials [10,11,12,13]. They are also economic and easy to produce [14]. Piezoresistive foams are composed of an insulating polymer matrix (polyurethane, polyethylene, melamine, etc.) and filled by a nanofiller (graphene, carbon nanotubes, conductive polymers, etc.) giving the foam electrical conductivity properties. The integration of nanofillers can be achieved by a simple coating method [10,13,15,16,17]. Piezoresistivity is a passive behavior based on the variation of electrical resistance of the foam to external loading. 

The characterization of the piezoresistive response of conductive foams in quasi-static regime was broadly investigated in numerous works [10,11,12,13,15,16,17,18,19,20]. However, dynamic characterization of such materials has been poorly studied especially for impact sensing [21]. This paper aims to evaluate the performance of an antistatic foam as a pressure/strain sensor made from conductive polyurethane on a large dynamic measurement range by identifying the most influent parameters.

In order to evaluate the sensing performances of the piezoresistive foam in quasi-static and dynamic regime, electrical sensitivity must be considered. In the scientific literature, there is no universal model quantifying the most influential parameters on the sensitivity of piezoresistive foams. The main objective of this article is to better understand the relation between mechanical and electrical behavior by identifying various parameters influencing sensitivity, such as the initial conditions (conductivity, mechanical environment) and foam viscoelasticity (compression velocity). Polymer-based foams are mainly viscoelastic materials, meaning that the elastic modulus is proportional to the strain rate. To the authors’ knowledge, there is a gap in research works about the influence of strain rate over sensitivity on large-scale measurements. A direct link between those parameters would permit an effective evaluation of the severity of the damage on a structure by the foam sensor. To do so, the foam microstructure is analyzed by optical microscopy, and samples are subjected to compressive tests with in situ conductivity measurement at various strain rates until 10 s^−1^. Low-energy impacts tests (<250 mJ) are conducted to assess the dynamic electrical response. Finally, the piezoresistive response of the embedded foam in a sandwich structure is investigated under low-energy impact (2 J).

This paper starts with a description of the materials and the elaboration of tested samples. Polyurethane-based foam filled with active carbon was studied. The method of measuring electrical resistance is described. Quasi-static tests are carried out in order to verify the responsiveness of the foam facing cyclic compressions. The basic electromechanical response of the foam is then analyzed. Attention is paid on the first compression analysis to compare the sensitivity of the electromechanical response of the foam when the sample is either in a free or confined configuration with a closed structure. The next part aims at identifying the influence of the material viscoelasticity and the initial conductivity on the detection performance by means of a dynamic mechanical analysis. The potential use of piezoresistive foams for SHM is then evaluated by analyzing the electrical response during low-energy impacts (2 J) inflicted on the sandwiched structure having the conductive foam in it. This research work ends with a conclusion and by giving some outlooks.

## 2. Materials and Methods

### 2.1. Material

The considered material was a very low density (18 ± 1 kg·m^−3^) polyurethane-based foam (PUF-AC) impregnated with activated carbon purchased from Industrie-Shaulm-Produkte GmbH & Co. KG, Limburg, Germany, under the product name of ELS-soft. The impregnation and fabrication methods are kept confidential by the manufacturer. The foam has a three-dimensional microstructure. It consists of set of open pores, which is an assembly of struts forming hexagons or pentagons of 300 μm side on average, randomly oriented in space as shown in Figure 1, obtained with the Keyence VHX-500 series from Keyence, Bois-Colombes, France, digital microscope at ESTACA’LAB, Laval, France. Thanks to its electrical conductivity and energy absorber capacity, this kind of foam is broadly used for antistatic packaging. 

### 2.2. Sample Elaboration and Determination of Electrical Resistance

The samples with square surfaces of 35 mm side by 12 mm thick were cut with a cutter from a 2 × 2 m plate. Depending on the position on the plate where the sample was cut, it had an intrinsic conductivity R_0_ varying from 1 to 10 kΩ. This dispersion of R_0_ is due to manufacturing process. Therefore, to obtain a homogeneous series of samples, they were cut in the same area (center) of the plate. 

In order to correctly measure the electrical resistance, electrodes made from a thin copper tape were placed on the top and bottom surfaces of the samples (Figure 2a). They covered 100% of their horizontal surface; nevertheless, the high porosity (>95%) and roughness (Figure 1c) of the foam make a good adhesion difficult between its surfaces and the electrodes, inducing unstable contact electrical resistance. 

Therefore, the top and bottom surfaces were initially covered by a thin layer of a highly conductive (silver content) varnish (LOCTITE^®^ 3863 Circuit+™) (Figure 2a). This varnish layer thus filled some of the hollow cells of the foam surface at 1 mm maximum penetration, covering less than 10% of the total thickness (Figure 2b). Resistance contact was identified in various research works on conductive foams; electrodes made from a dense silver paint seems to be a popular solution [12,15,21].

A low direct current (30.8 µA) from a low-current generator was sent to PUF-AC samples via wires that were connected to a laboratory-made data acquisition system (DAQ) for measuring the tension in a 0–10 V voltage range with an accuracy of 15 effective bits (Figure 3). The electrical resistance was determined with Ohm’s law, considering the foam as a series of 3 variable resistances depending on mechanical parameters. Electrodes made from silver varnish assure good electrical contact, so the terms *R_contact1_* and *R_contact2_* can be nullified in Equation (1).
*R_total_* = *R_contact1_* + *R_foam_* + *R_contact2_*(1)

### 2.3. Quasi-Static Tests: Method

The electromechanical behavior of PUF-AC was first characterized with cyclic uniaxial compressions in two initial configurations (Figure 4) using a conventional testing machine, model INSTRON 3369, with a cell force of 500 N based at ESTACA’LAB. In a free configuration, PUF-AC samples were placed between two plates re-covered with Teflon (for electrical insulation) and vertically compressed at 80% of the initial thickness with the testing machine. A 2.5 kS/s data acquisition rate option acquired load and strain measurements simultaneously. The testing machine was operated in displacement control. A PUF-AC sensor for SHM would be likely preloaded when embedded in a structure so its performances are supposed to be modified; therefore, a second configuration was studied. PUF-AC samples were inserted in a structure designed to assure lateral pre-loading. This structure was a cuboid cavity of 34 mm width that was pressing against the lateral surfaces of the foam sample (Figure 4b). In both configurations, electrical data (tension) were simultaneously acquired with mechanical data (force, strain). PUF-AC samples were compressed during 10 cycles at a compression velocity of 8 mm/min in relation to the sample thickness. 

The evolution of conductivity is quantified by the variation of the relative electrical resistance R expressed as (2):(2)$$\frac{\Delta R}{R_{0}}(t)=\frac{{(\;R(t)\;-\;R}_{0}\;)}{R_{0}}\;$$

*R* (Ω): resistance at t time (s);

*R*_0_ (Ω): resistance at *t* = 0 s. 

PUF-AC sensor performance is evaluated through its electrical sensitivity in relation to strain *S_ε_* (3) and stress *S_σ_* (kPa^−1^) (4).
(3)$$S_{\varepsilon }=d\left({\Delta R/R}_{0}\right)\;∕\;d\varepsilon \;$$
(4)$$S_{\sigma }=d\left({\Delta R/R}_{0}\right)∕d\sigma \;$$

Stress on PUF-AC sample is quantified from the force *F(t)* measured by the testing machine in relation with the sample surface S considered as constant (6).
(5)$$\sigma (t)=\frac{F\left(t\right)}{S}\;$$

*σ*(*t*) (kPa): stress at t time;

*F*(*t*) (N): force at t time;

*S* (mm²): surface of foam sample (constant). 

### 2.4. Strain Rate Dependancy Test Method 

In order to better understand the dependency of the electromechanical behavior on the strain rate, the foam samples were compressed at 80% of their thickness at various strain rates (10^−2^, 10^−1^, 1, 10 s^−1^), in a free configuration on a DMA machine (E3000, ElectroPuls^®^ Instron 3 kN, capable of dynamic cycle load up to 100 Hz, Instron, High Wycombe, UK) in displacement control mode. 

Strain rate is defined as the displacement velocity of the superior compression plate of the E3000 machine (Figure 5) in relation to the sample thickness (*e*). It is simply the temporal derived expression of Equation (5):(6)$$\dot{\varepsilon }=\frac{d\varepsilon \left(t\right)}{\mathrm{dt}}=\frac{d}{\mathrm{dt}}\left(\frac{\;H\left(t\right)\;{-\;H}_{0\;}}{e}\right)\;$$

*H*(*t*) (mm): position of top plate at *t* time;

*H*_0_ (mm): position of top plate at *t* = 0 *s*;

*e* (mm): sample thickness.

Strain rate $\dot{\varepsilon }$ (s^−1^) is a test consign, so it is considered as a constant during the entire test. 

Compressive strain and loading data were acquired via an LVDT position sensor of the E3000 machine and a load cell of 500 N capacity with accuracy of 10^−3^ N. Electrical resistance was obtained via the same method as explained in Figure 3.
Figure 5Experimental setup on E3000.
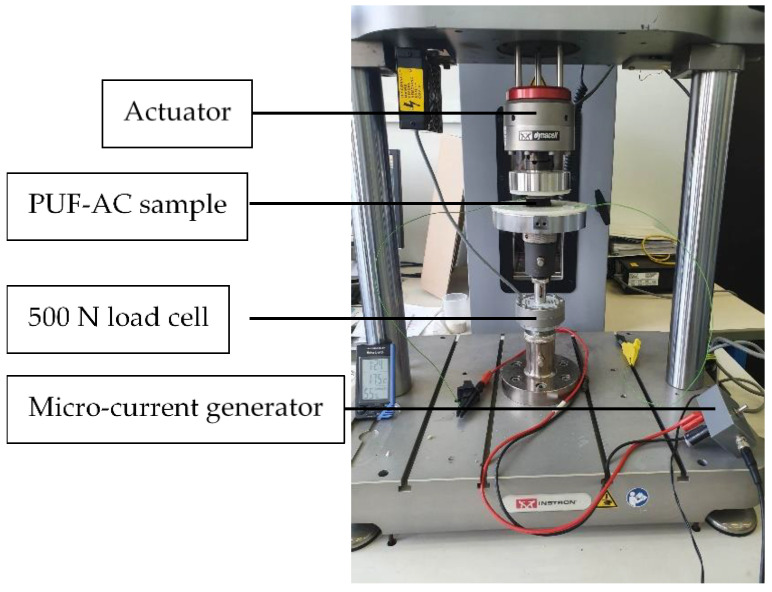


### 2.5. Assessment of the PUF-AC Piezoresistive Response under Impact Test

The dynamic response of PUF-AC was analyzed by subjecting it low-velocity impacts (<10 m/s). For this purpose, an impactor in Teflon of 16 cm long by a square surface of 36 mm wide was dropped vertically (Figure 6) at various heights ranging from 1 to 16 cm (Table 1). The surface dimensions of the impactor were chosen to assure that the sample undergoes the most planar compression as possible. In this test, impact energy is considered as equal to potential energy and is defined as:(7)Eimpact=Epotential=mgh

*M* (kg): mass of impactor (=145 g); 

*g*: gravity acceleration on earth (9.81 m·s^−2^);

*h*: height of drop (m).

Electrical resistance of samples is recorded with the same method as was used in the quasi-static tests.
Figure 6A 3D illustration of impact test setup.
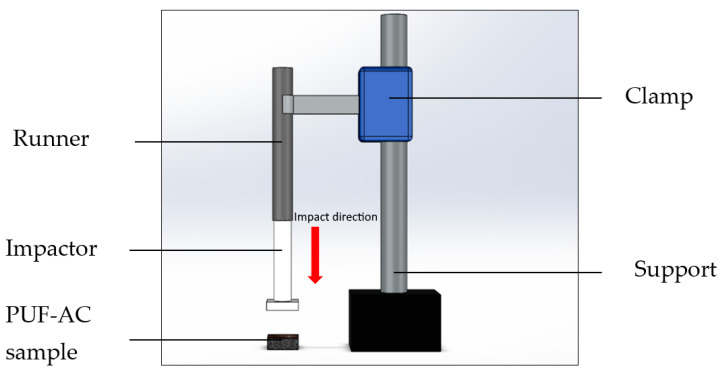

sensors-23-05161-t001_Table 1Table 1Impact energy absorbed by PUF-AC samples with the corresponding drop height.E_potential_ (J)Drop Height (cm)0.01410.02820.05740.11480.171120.22716

### 2.6. Low-Velocity Impacts on Smart Structure: Test Method

To demonstrate the potential use of PUF-AC for SHM application, it was first necessary to test the foam response when integrated into a structure. A sandwich composite structure was chosen. The core of the sandwich was a block that was 30 mm thick by 105 × 105 mm wide and made of a rigid plastic (PVC) foam (80 kg·m^−3^, AIREX C70.80^®^, Airex AG, Sins, Switzerland). Moreover, a cavity 34 mm wide and 11 mm deep was created from the center of the top surface of the core where the PUF-AC was placed. The PUF-AC sample was precompressed by 1 mm of its height and width to avoid problems of contact discontinuity between the foam and the structure. Two skins of 2 mm thick made of stratified composite (epoxy filled with 16 unidirectional [0°; 90°] carbon fibers layers) were placed on the top and bottom of the core. The two skins were attached to the core via 4 screws. 

Impact tests were conducted with a IMS10^®^ drop weight impact tester (from IMATEK, Knebworth, UK) equipped with a pneumatic break to avoid rebounds. An 8 kg and 20 mm diameter steel hemispherical impactor of was dropped from a 25.48 mm height on the top skin. The sandwich structure then underwent an impact energy of 2 J (Equation (7)) corresponding to an impact velocity of 0.7 m·s^−1^. The impact point was directly facing the symmetric center of the top surface of the PUF-AC foam (Figure 7).

Mechanical data (force, position) were acquired from a laser position sensor (20 kHz, 10 μm precision) and a cell force (20 kHz, 30 kN capacity, ±5 N accuracy) placed on the impactor. The same method of resistance measure was used with a higher sample rates of 100 kHz. All the signals were acquired on a chain of 0–10 V range with an accuracy of 16 bits. Moreover, they all were triggered at the same time from a defined position of the impactor before collision.

## 3. Results and Discussion

### 3.1. Piezoresistive Response and Conductive Mechanism of PUF-AC

Initially, it is essential to analyze the electromechanical behavior of PUF-AC during the first compression in free configuration. On the stress/strain curve in Figure 8a, an elastoplastic behavior is observed. It is a typical mechanical response of polymeric foams, which is described by three stages: (E) elasticity (ε = 0–10%); (P) plateau (ε = 10–55%); and (D) densification (ε = 55–80%). Figure 9 shows the deformation of a cell at each compressive phase. This microcell pictured in Figure 9 is situated on the thickness face of the sample.

The conduction mechanism is initially based on the intrinsic resistance of the filler particles forming a conductive network. Two forms of conductance are operated in the network: tunneling effect and ohmic conductance (Figure 10). Depending on the interparticle distance, the tunneling effect is the probability of an electron passing from one filler particle to another through a thin layer of insulating material (polyurethane matrix). This quantic effect is directly related to the concentration and repartition of the conductive filler into the polyurethane matrix. A higher filler concentration ratio increases the number of contact points between filler particles, thereby reducing the influence of tunnel resistance [10]. 

During the elastic phase, the stress increases linearly until reaching a “plateau constraint” of 3.5 kPa approximately (Figure 8a). Mechanical models [22] describe this phase as a linear bending of the cells struts of the foam (considered as beams). In the plateau phase, a low augmentation of stress is observed until *ε_densification_*. This is due to struts buckling, collapsing cells with low effort (Figure 9b). It is suggested that the increasing resistance ratio observed from *ε*_0_ to *ε_densification_* (Figure 8b,c) is due to progressive reduction in the influence of the tunneling effect; during this phase, the distance between filler particles is increasing until reaching the limit of tunnel distance. Moreover, compression damages induced on the microstructure perturbed the conduction network (red circles in Figure 9c,d).

The densification phase starts at approximately 55% of the compressive strain. It corresponds to the threshold at which the foam cells collapse (Figure 9c,d). The mechanical response shows an increase in stiffness (Figure 8a). From an electrical point of view, the conductive network is denser, which reduces foam resistance (Figure 8c and Figure 9c). Regarding Figure 9d, the struts of the cell are in contact, providing ohmic conductance, which is a result of a decrease in resistance. These observations on conductive mechanism have been well referenced [23,24].

After being completely compressed, the sample foam is supposed to be damaged. Partial or complete rupture of struts can be observed (Figure 9c). Consequently, the conductive network is also damaged. Moreover, the initial conductivity R_0_ will never be recovered after unloading. 

As represented in Figure 8b,c, resistance ratio curve is divided accordingly into three deformation phases. Using the least mean square method, each domain is approximated by a linear curve with a slope corresponding to *S_ε_* (Figure 8b) and *S_σ_* (Figure 8c).

### 3.2. Cyclic Compressions: Electromechanical Hysteresis

The cyclic compressions test, performed with Instron 3369, aimed to study the repeatability of the electromechanical response of PUF-AC. 

Cycle 1: The mechanical behavior during cycle 1 of compression is described by a hysteresis curve on graph in Figure 11a. The diminution of stress level during unloading phase was due to the Mullins effect, which is directly related to the viscoelasticity property of PUF-AC. The evolution of relative resistance was also nonrepeatable during unloading phase (Figure 11b,c). Little information in the scientific literature is reported to explain this observation. This work suggests that the main consequence is linked to viscoelasticity and cells being damaging. Viscoelasticity can be described as a linear increase of the rigidity of the studied material caused by the augmentation of the strain rate during the elastic loading phase. The Maxwell model represented by association of spring and dashpot describes viscoelastic behavior of materials [25]. Under loading, molecular chains of polyurethane were temporarily in movement by creating entanglements [26]. This mobility canceled the tunneling effect because the tunnel distance between particles was broken. In consequence, the main conduction mechanism in operation was ohmic conductance, which depends on the cell contacts variations. When the sample returned to its initial state after a certain recovering time, the two conduction mechanisms (tunneling and ohmic conductance) operated again. The similar form of curves was observed after one day at rest. However, the general resistance level was increased because of irreversible damages caused by compressions (Figure 9c,d). 

Cycle 2 to 10: At the second compression cycle, an electromechanical hysteresis was observed. The relative resistance of PUF-AC sample had increased by 160% compared to the starting first cycle. As explained previously, at this time of experiment, the evolution of foam conductivity only depends on ohmic conductance, so the progressive diminution of relative resistance during loading phase (Figure 11b–d) was justified. From 2nd to 10th cycle, the hysteresis effect progressively decreased, and the electromechanical behavior tended to be more stable (Figure 11). Along with compressive cycles, the polyurethane chains and filler particles were moving with the same inertia without recovery time, which could explain this form of stability regarding the hysteresis behavior.

Tests in fatigue and a complete history of foam events would permit a better interpretation of its electrical response.

### 3.3. Influence of Pre-Stress Condition on Performances

As explained in previous parts, PUF-AC was chosen to be integrated in a structure. The foam samples would be likely pre-stressed by its implementation in a structure. The experiment aims to understand the influence of a lateral pre-stress condition on its electromechanical behavior and performances.

The stress–strain curve in Figure 12a shows that samples with the pre-stress condition have an elastoplastic behavior with more stiffness. An increase in plateau constraint (+ 0.5 kPa) can be observed. The piezoresistive response of confined samples is broadly similar to samples in free configuration. Nevertheless, the results show a lower electrical sensitivity to strain (*S_ε_*) and stress (*S_σ_*) in the elasticity phase (see Table 2), with a relative difference of 58% and 60%, respectively.

The electrical sensitivity would likely be diminished by the implementation of PUF-AC in a structure. Moreover, the performances of the PUF-AC (reported in Table 2) are comparable to other conductive PU foams [10] studied in the literature.

### 3.4. Strain-Rate Dependency

Figure 13a describes the electromechanical behavior of samples. The three phases of elastoplastic behavior are still visible (at the same strain levels), but the global level of stress proportionally increases with the strain rate, confirming the rate dependency of polymeric foams. Moreover, it can be observed that the stiffness is superior regarding the progressive augmentation of Young’s modulus in Table 3. At 10 s^−1^, the stress–strain curve in the plateau phase is not linear and presents a decrease at *ε* = 30%. This loss of rigidity is possibly due to an eventual weakening of foam samples.

Figure 13 shows the potential relation between viscoelasticity and electrical sensitivity in the PUF-AC sample. Regarding Figure 13d, all sets of the tested samples have a homogeneous initial resistance, meaning that R_0_ can be considered as a controlled parameter. 

Elasticity and plateau phases: The progressive reduction in (Δ*R/R_0_)_max_* and *S_ε_* are proportional to the augmentation of the strain rate (see Table 3). It is supposed to be due to the augmentation of stiffness induced by viscoelasticity, which lowers tunnel resistance; nevertheless, at sample scale, it is difficult to confirm this interpretation. Numerical simulation of foam deformation at local scale (cell) would help with understanding this result. Figure 13b shows a similar electrical response and sensitivity (*S_ε_*) between tests at 1 and 10 s^−1^, suggesting the electrical sensitivity is more influential at lower strain rates. Regarding Figure 13c, by considering compressive stress, electrical sensitivity (*S_σ_*) is clearly inferior between tests at 1 and 10 s^−1^ strain rates with a relative difference of 75%. Until densification, viscoelasticity seems to be influential on detection performances of the foam. 

Densification phase: results from Table 3 and Figure 13 show a noninfluence of strain rates on electrical sensitivity (*S_ε_*, *S_σ_*) except for *S_σ_* at 10 s^−1^. This exception is due to an eventual damaging of samples at 10 s^−1^. From these results, it can be concluded that viscoelasticity does not influence foam sensitivity much during large deformations when [*ε ≥ ε_densification_*].

In summary, a distinct variation of electromechanical sensitivity of PUF-AC is observed with the augmentation of strain velocity until the densification. This result is interesting because it shows the ability of the foam to detect various loading velocities, which represents a potential advantage for impact severity evaluation.

### 3.5. Electromechanical Parameter 

As mentioned in the Material and Methods section, the resulting foam conductivity is dependent of the manufacturing process. Initial resistance is directly related to filler concentration. In order to quantify *R_0_* influence on sensitivity, PUF-AC samples with different conductivities were tested with the exact same method as strain dependency test. From Table 2 and Table 3, the maximum relative resistance (Δ*R/R*_0_*)_max_* obtained during the first compression is a great indicator for sensitivity. 

Figure 14 shows the piezoresistivity response as a function of *R*_0_ for different strain rates $\dot{\varepsilon }$. Each ensemble of ((Δ*R/R*_0_)_*max*_; *R*_0_) points are approximated as linear curves (Figure 14a). Moreover, when the *R*_0_ value approaches zero, conductivity tends towards infinity, meaning that the considered material is an electrical conductor. Therefore, (Δ*R/R*_0_)*_max_* can be considered as negligible when *R*_0_ = 0 because electrical conductors, such as metals, are faintly sensible to conductivity variation when subjected to loads. Each slope of the linear curve is considered as a new measurement parameter that is arbitrary named electromechanical parameter (*P_EM_*) for the evaluation of electrical sensitivity to strain rate (Figure 14b), considering *R*_0_. It is expressed by Equation (8).
(8)$${\left(\Delta R/R_{0}\right)}_{\max}{=P}_{\mathrm{EM}}{\;\times \;R}_{0}\;$$

Firstly, results from Figure 14a show that for a fixed strain rate, when *R*_0_ is increasing, (Δ*R/R*_0_)*_max_* is augmented, confirming the augmentation of sensitivity with *R*_0_. Secondly, referring to linear curves, for a fixed *R*_0_ value, (Δ*R/R*_0_)*_max_* is superior proportionally to strain rate. Eventually, it can be clearly observed (Figure 14b) that sensitivity to load velocity is more important when PUF-AC conductivity is lower. *P_EM_* can be approached with a logarithmic expression of strain rate (9).
(9)$$P_{\mathrm{EM}}=\left(-0.008\right)\;\times \;\ln(\dot{\varepsilon })+0.067$$

The resulting Equation (10) can be expressed as below:(10)$${\left(\Delta R/R_{0}\right)}_{\max}=[\left(-0.008\right)\;\times \;\ln(\dot{\varepsilon })+0.067{]\;\times \;R}_{0}$$

### 3.6. Piezoresistivity Behavior during Dynamic Loading

This subsection aims to present the characterization of dynamic behavior of the PUF-AC subjected to light shocks. To the authors’ knowledge, only one publication reports about impact sensing of piezoresistive foams [21].

Unlike quasi-static tests, samples undergo a quick compression during impact. The foam is acting as a soft spring: a part of the impact energy is absorbed by the foam and is partly redistributed to the impactor until it is immobilized on the sample, causing a relaxation condition. 

The dynamic behavior of PUF-AC observed in Figure 15a is quite different to quasi-static response. It can be broadly described by a steep augmentation of relative resistance during impact followed by a smooth decrease until reaching a stable of resistance level. This form of electrical response is similar to research work of Boland et al. [19] (*E_impact_* ≤ 5 mJ) on PU foam filled with graphene.

However, for an impact energy superior to 114 mJ (*h* > 8 cm), an abrupt diminution of resistance (≈−10%) systematically precedes the impulsion (Figure 15a). From a physical point of view, this fact can be connected to the quasi-complete compression of the sample (*ε* ≈ 100%, densification phase), which is not reached at lower impact energies. The distance between electrodes is clearly reduced (capacitive effect), resulting in a better conductivity. From this suggestion, the first resistance impulsion is attributed to the foam response: the impactor is rebounding on the sample. It is similar to the hysteresis effect observed in quasi-static tests, where resistance increases during unloading phase of compression (Figure 11b). Moreover, after reaching (Δ*R/R*_0_)*_max_*, another rapid diminution of resistance is observed. This behavior is due to the rebound of the impactor on the sample until reaching complete energy absorption.

In order to compare the sensitivity of PUF-AC in relation to impact energy, the maximum reached resistance (Δ*R/R*_0_)*_max_* is presented as a function of *E_impact_*. A good repeatability can be observed, and a relation between those parameters is expressed as a power law (11):(11)$${\left(\Delta {R/R}_{0}\right)}_{\max}=1.8872{\left(E_{\mathrm{impact}}\right)}^{0.7024}$$

### 3.7. Low-Velocity Impact on Smart Structure: Results

The electromechanical response of the smart composite structure (SCS) is reported in Figure 16. The impact event analysis is divided in three main times: *t*_1_ (=15.5 ms) is the time at which the hemispherical surface of the impactor reaches the top surface of the SCS (vertical position = 0 mm), with an initiation of the contact force.*t*_2_ (=22.4 ms) is the time at which all the kinetic energy of the impactor has been transferred to the SCS. The impact indentation reaches 3.6 mm for a maximum force of about 400 N. The energy is then dissipated through longitudinal and transverse waves in the sandwich structure [27]. The PUF-AC detects the impact with a delay corresponding to ${\Delta t=t}_{2\;}-{\;t}_{1}\;$. A first piezoresistive response is observable at *t*_2_ first with a negative peak of the relative resistance (−30%) followed by a wave form of the relative resistance curve with a maximum amplitude of −109%.*t*_3_ (=31.5 ms) is corresponding to the loss of contact with impactor at the end of the rebound phase [$t_{2}$, $t_{3}$]. The PUF-AC final and initial conductivities are similar.

The first passive sensing of impact has been demonstrated by the PUF-AC embedded in the SCS. The absence of significant effects related to the tunnel resistance during the loading phase is due to higher velocity impact and the boundary conditions (lateral and vertical pre-compression) according to the results presented in Section 3.3 and Section 3.4.

## 4. Conclusions

To conclude, the aim of this work was to characterize the piezoresistive behavior of an antistatic foam in quasi-static and dynamic regimes in order to evaluate its sensing performances and its ability to detect an impact when embedded in a structure. The polyurethane foam filled with active carbon (PUF-AC) shows a great responsiveness during the first compression test with an important electrical sensitivity at low strain (*S_ε_ =* 1.46). The repeatability of this response is discussed as the conduction mechanism varies starting from the unloading phase of first compression due to polymer viscosity and foam damages. However, from cycle 2 to 10, a repeatable electromechanical hysteresis is then observed, which tends to minimize with the augmentation of compression cycles.

As the performances of PUF-AC *(S_ε_, S_σ_)* in quasi-static regime are comparable to several reference works, the fabrication parameters of the foam are not considered in the study. A special attention is paid to key parameters such as boundary conditions, viscoelasticity, and initial conductivity. The pre-stress condition is considered as a new influence parameter on sensitivity (maximum relative difference of 32 % for *S_ε_*). The influence of viscoelasticity is studied by loading the PUF-AC at different compression velocities. A non-negligible dependency of sensitivity to strain rate ($\dot{\varepsilon }\in {\;[10}^{-2}{\;;\;10\;s}^{-1}]$) is clearly observable with a maximum decrease of 40% *(S_ε_).* In addition, a new empirical parameter *(P_EM_)* has been proposed to describe the dependency of sensitivity to viscoelasticity and conductivity; the variations of sensitivity due to strain rate are amplified with a higher foam initial conductivity.

The dynamic characterization of the PUF-AC was investigated. The impact tests directly applied on PUF-AC sample show a great detection capacity on large deformations (*ε* ∈ [0; 80%]) with a good sensitivity (Δ*R/R*_0_*)_max_* > 60%) to low impacts (*E_impact_* ≤ 227 mJ). 

Moreover, a demonstrative experiment of impact detection by PUF-AC embedded in a sandwich structure in composites has been conducted. These results constitute a first step for the feasibility of SHM applications integrating piezoresistive foam sensors.

Even if the PUF-AC showed its ability to detect a transient loading on a structure, its electrical response needs to be deeply investigated by numerical simulations and additional impact tests on Hopkinson bars. In addition, the durability of the material should be studied in varying atmospheric conditions (humidity, temperature).

## Figures and Tables

**Figure 1 sensors-23-05161-f001:**
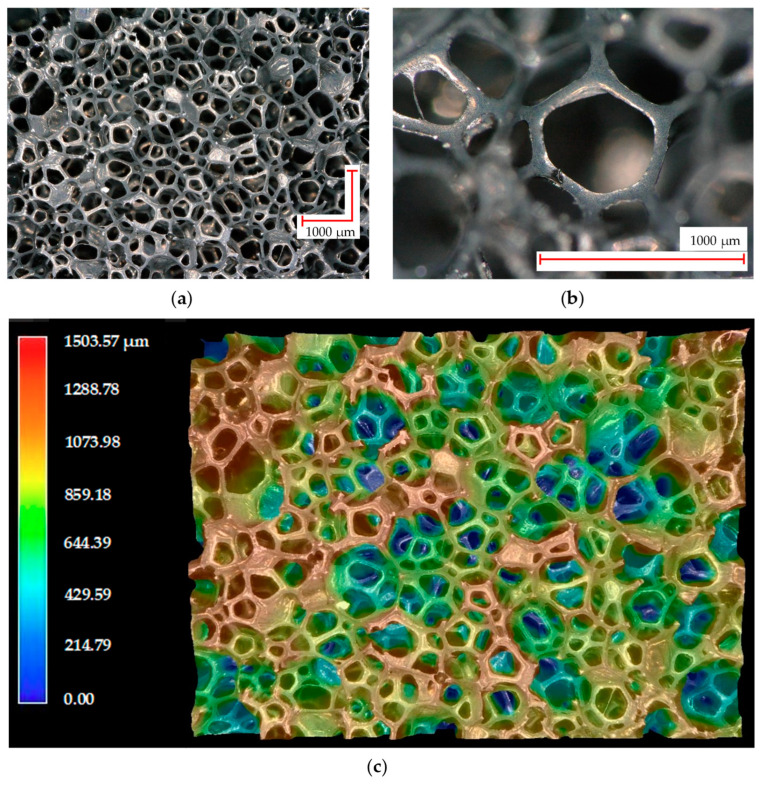
Optical microscopy photographs of the PUF-AC microstructure: (**a**) on surface; (**b**) at cell scale; and (**c**) with depth representation.

**Figure 2 sensors-23-05161-f002:**
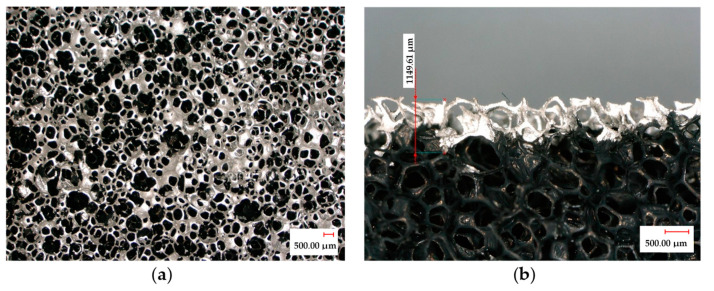
Optical microscopy photographs of the PUF-AC microstructure covered by varnish layer: (**a**) compression surface view; (**b**) thickness view.

**Figure 3 sensors-23-05161-f003:**
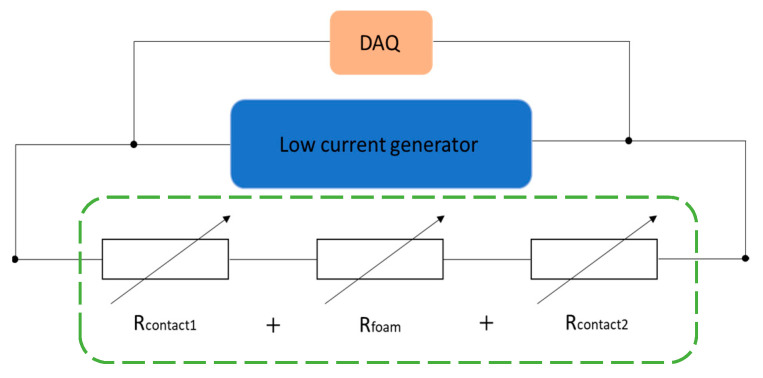
Electrical illustration of PUF-AC foam and its electrical measure setup.

**Figure 4 sensors-23-05161-f004:**
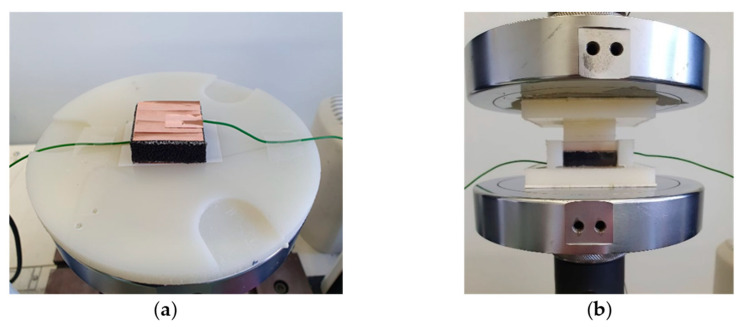
PUF-AC sample in free configuration (**a**) and confined configuration (**b**).

**Figure 7 sensors-23-05161-f007:**
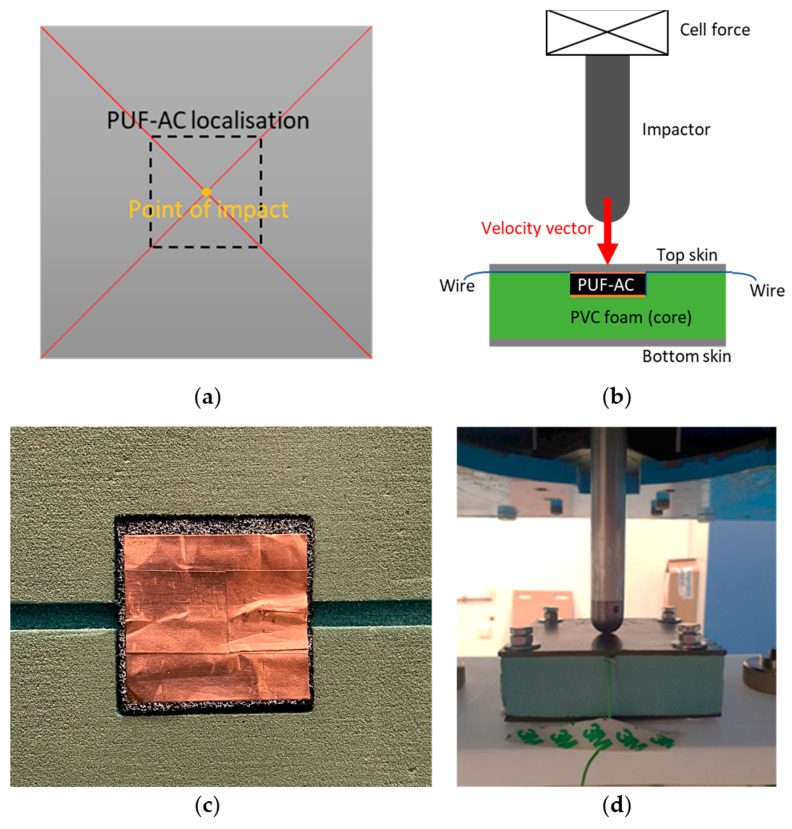
Schemes and photograph of impact test on smart sandwich a structure (SCS): (**a**) schematic top view of SCS with localization of impact; (**b**) schematic front view of SCS with description of elements of the impact test; (**c**) photograph of the embedded PUF-AC sample in the sandwich core; and (**d**) photograph of the SCS and impactor.

**Figure 8 sensors-23-05161-f008:**
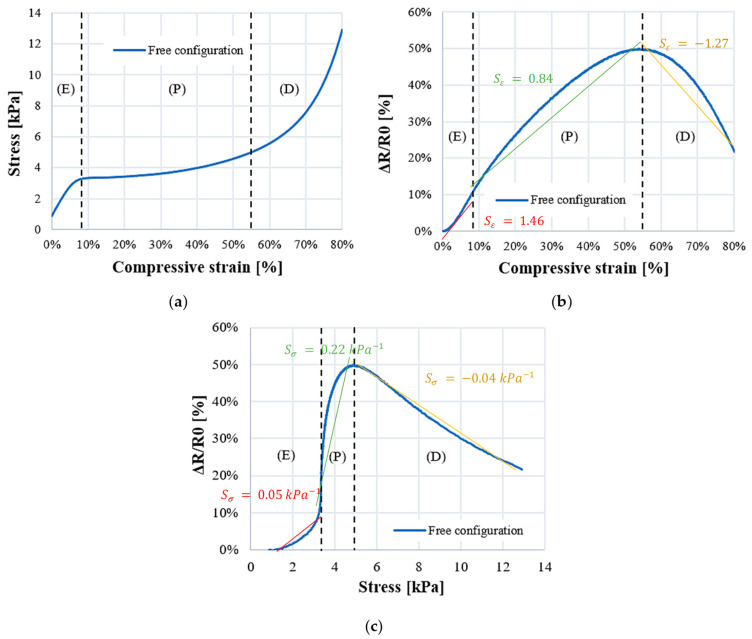
PUF electromechanical behavior during first compression in free configuration: (**a**) compressive stress-strain curve; (**b**) relative resistance as a function of compressive strain; and (**c**) stress.

**Figure 9 sensors-23-05161-f009:**
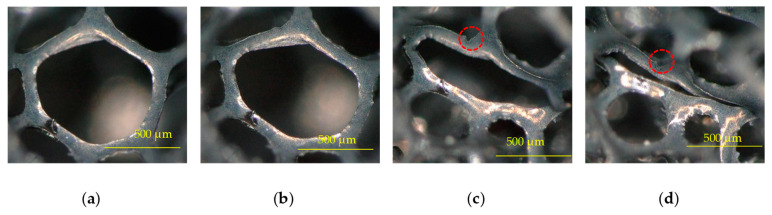
Foam cell deformation during compression: (**a**) *ε*_0_ = 0; (**b**) *ε_elasticity_* = 10%; (**c**) *ε_densification_* = 55%; and (**d**) *ε_final_* = 80%.

**Figure 10 sensors-23-05161-f010:**
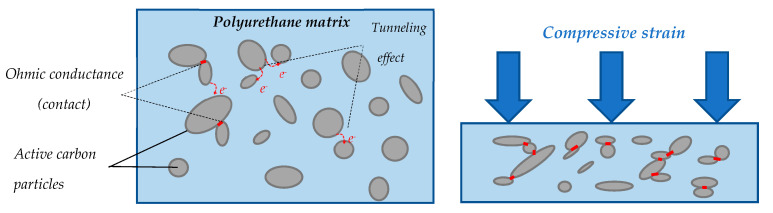
Schematic representation of the conductive mechanism.

**Figure 11 sensors-23-05161-f011:**
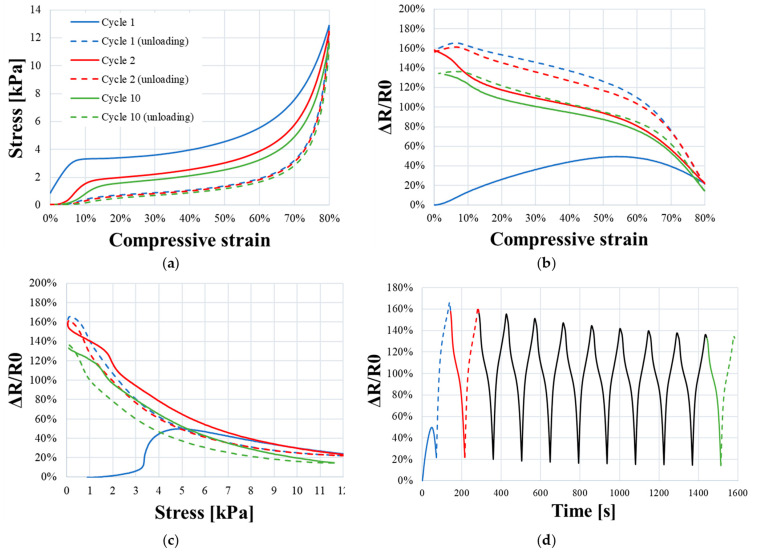
Evolution of the relative resistance of the piezoelectric foam during cyclic compressions over ten cycles: (**a**) compressive stress strain curve; relative resistance as a function of compressive (**b**) strain, (**c**) stress, and (**d**) time.

**Figure 12 sensors-23-05161-f012:**
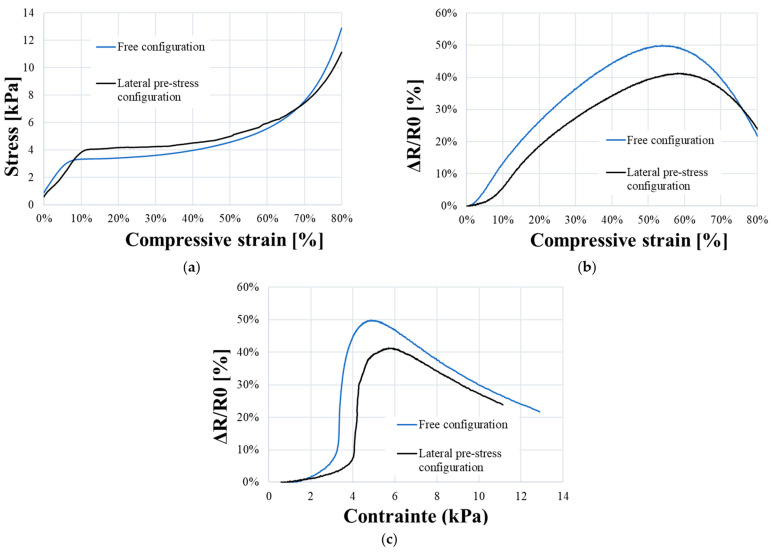
Comparison of electromechanical behavior of PUF-AC samples in a free and pre-stress configuration: (**a**) compressive stress–strain curve; relative resistance as a function of compressive (**b**) strain and (**c**) stress.

**Figure 13 sensors-23-05161-f013:**
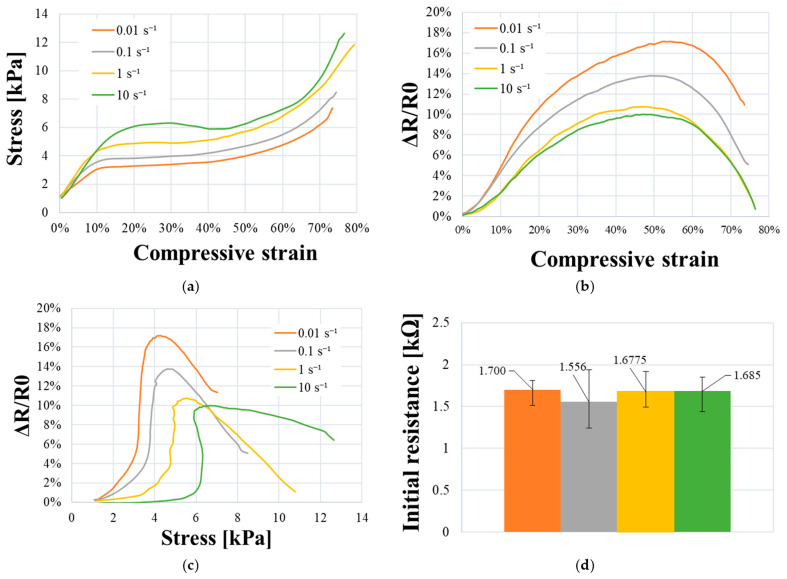
Electromechanical characterization of PUF-AC at various strain rates with: (**a**) stress–strain curve; relative resistance as a function of (**b**) compressive strain and (**c**) stress; and (**d**) initial resistance average of each samples set.

**Figure 14 sensors-23-05161-f014:**
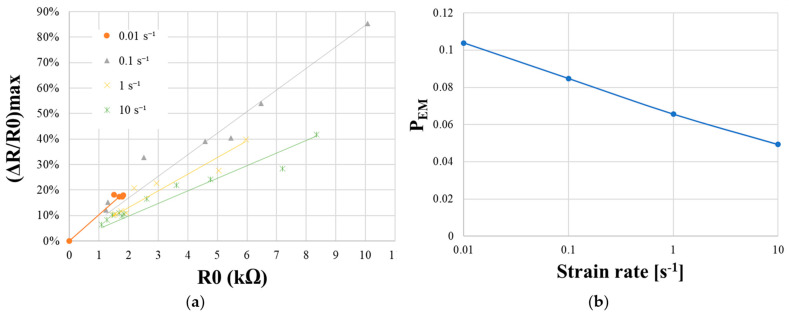
Analysis of electromechanical parameter *(P_EM_)*: (**a**) (Δ*R/R*_0_)*_max_* represented as a function of *R_0_* for different strain rates; (**b**) *P_EM_* as a function of strain rate in logarithm scale.

**Figure 15 sensors-23-05161-f015:**
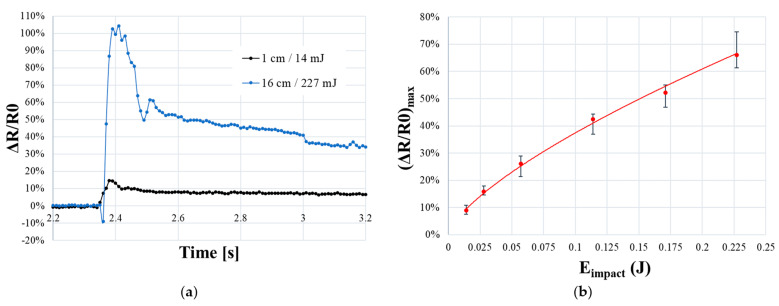
Electrical response of PUF-AC during light impacts: (**a**) temporal response of relative resistance at the minimum (14 mJ) and maximum (227 mJ) impact energies; (**b**) maximum resistance ratio as a function of impact energy.

**Figure 16 sensors-23-05161-f016:**
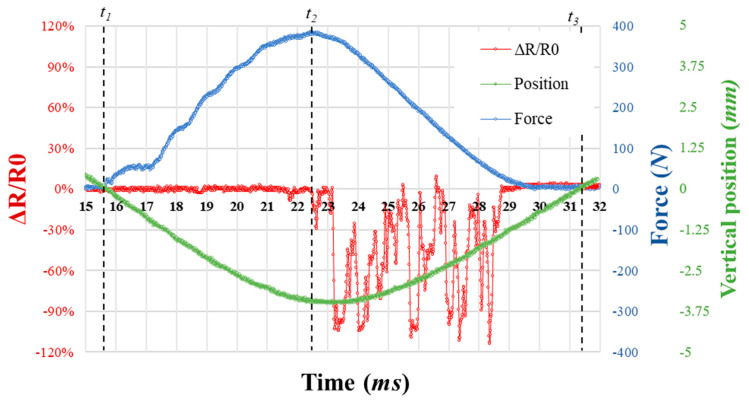
History of 2 J impact on the SCS with the time evolution of force during collision (blue curve); the vertical position of the impactor (green curve); and the electrical response of the embedded PUF-AC (red curve).

**Table 2 sensors-23-05161-t002:** Comparison of electrical data between free and pre-stress configuration.

Samples	*R*_0_ [kΩ]	(Δ*R/R*_0_)*_max_*	Elasticity		Plateau		Densification	
			*S_ε_*	*S_σ_*	*S_ε_*	*S_σ_*	*S_ε_*	*S_σ_*
Free	5.81	49.63%	1.46	0.05 kPa^−1^	0.84	0.22 kPa^−1^	−1.27	0.04 kPa^−1^
Pre-stress	5.71	41.35%	0.62	0.02 kPa^−1^	0.71	0.19 kPa^−1^	−0.86	0.03 kPa^−1^

**Table 3 sensors-23-05161-t003:** Electromechanical response data obtained at every strain rate for the PUF-AC foam.

Strain Rate [s^−1^]	Young’s Modulus [kPa]	(Δ*R/R*)*_max_*	Elasticity		Plateau		Densification	
			*S_ε_*	*S_σ_*	*S_ε_*	*S_σ_*	*S_ε_*	*S_σ_*
0.01	19.63	17.17%	0.487	0.025 kPa^−1^	0.223	0.077 kPa^−1^	−0.411	−0.026 kPa^−1^
0.1	22.881	13.89%	0.446	0.019 kPa^−1^	0.211	0.074 kPa^−1^	−0.454	−0.026 kPa^−1^
1	31.058	10.76%	0.334	0.008 kPa^−1^	0.141	0.037 kPa^−1^	−0.436	−0.02 kPa^−1^
10	32.35	9.97%	0.287	0.002 kPa^−1^	0.112	0.044 kPa^−1^	−0.413	−0.005 kPa^−1^

## Data Availability

The data presented in this study are available upon request from the corresponding author.

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
