# Peer review of "Piezoresistive Behavior of a Conductive Polyurethane Based-Foam for Real-Time Structural Monitoring"

_sensors, 2023, doi:10.3390/s23115161_

Round 1

Reviewer 1 Report

The article is very well designed. The authors need to address the following points to improve the quality of the article.

·         The abstract is a summary of the introduction, materials and method, results, and conclusion. This order needs to be followed. The methodology, results (quantifying data), and conclusion component of the abstract should be properly captured.

·         Cite table 1, in the explanation.

·         Correlate figures 8 and 9 properly. It helps readers.

·         “described by a 245 hysteresis curve on graph in Figure 9a.” Is it Figure 9a or 10a? Check the entire manuscript for correct citation of figures.

·         Kindly reconcile the conclusion with the study objectives.

·         What are the practical implications of this study and the future directions? kindly state?

·         The article must undergo language proofreading, as I found typos and grammatical errors.

 The article must undergo language proofreading, as I found typos and grammatical errors.

Reviewer 2 Report

This manuscript studied the piezoresistive behavior of a conductive polyurethane based foam for real time structural monitoring. Overall, this topic is interesting and meaningful, and the manuscript is also logical for reading. However, there are some concerns and suggestions which, if addressed, would enhance the quality of the manuscript. These are provided below:

1. In Abstract, “PolyUrethane foam filled with active carbon, namely PUF-AC, are manufactured and tested”, however, the PUF-AC was purchased other than manufactured by the authors!

2. The active carbon in PUF-AC is very important for their piezoresistive behavior, the authors should explore its amount and distribution in PUF rather than only rely on the information from the manufacturer. Otherwise, this manuscript is like an experimental report rather than a scientific research paper.

3. The ruler in Figure 1(a) and (b) should be revised clearly.

4. The authors mentioned “structural health monitoring” many times in Title, Abstract, Keywords,,, However, there is no practical applications examples in manuscript.

Reviewer 3 Report

In the manuscript of sensors-2403450, the authors aimed to study the potential use of the piezoresistivity effect of a conductive foam made of a flexible polyurethane matrix filled with active carbon for integrated Structural Health Monitoring (SHM) and damage assessment applications. However, the content and logic of research are not satisfied, so the manuscript should be modified by the following comments.

1. In section of introduction, the novelty of this research should be clarified deeply.

2. Offer a performance comparison table with other typical pressure sensors of conductive foam in detail.

3. In the section of Materials and Methods, the detailed fabrication methods of conductive foam should be added, inclunding the detailed information of activated carbon for particle size distribution.

4. Considering the adhesion problem of activated carbon on the PUF, the durability of the sensor needs to be further discussed and the current repeatability test is not sufficient to prove the durability.

5. The applications of real time structural monitoring are missing, or it should be added in the manuscript, or delete the relative term from the title.

6. The quality of Figure 10~14 should be enhanced, in which the letter may be adjusted to be larger.

7. The working mechanism of sensors needs to be discussed in detail, the following references may be helpful.

DOI: 10.1016/j.snb.2022.131497

DOI: 10.1016/j.cej.2020.124045

Moderate editing of English language is required.

Reviewer 4 Report

This paper investigates the use of a conductive foam made of flexible polyurethane filled with active carbon for integrated Structural Health Monitoring and damage assessment applications. The study includes the manufacturing and testing PolyUrethane Foam filled with Active Carbon (PUF-AC) under quasi-static compression tests and under DMA with in-situ electrical measurements. Additionally, a demonstrative experiment of a PUF-AC sample subjected to low-velocity impact assesses the potential of this type of material for damage assessments.

While the manuscript is well-organized and presents interesting results that are suitable for publication, it is important for the authors to note that several similar papers are already published in the literature. Therefore, the authors need to better clarify their original contribution to the field. To improve the quality of the research prior to publication, it is recommended that the authors conduct a more comprehensive characterization of the commercial activated carbon. This will enable a better understanding of the sensing mechanism, particularly with regard to the dimension, concentration, and impregnation method, which can greatly impact the effectiveness of the composite foam. Additionally, it is essential to investigate the reproducibility and durability of these materials to determine their potential for use in real-world conditions, as opposed to solely controlled laboratory environments.

I am not qualified to assess the quality of English in this paper

Round 2

Reviewer 1 Report

The authors have addressed all the queries. The article may be accepted in its present form. 

Reviewer 3 Report

The authors have well addressed all my comments, so I recommend it to be accepted for publication in present form.

Reviewer 4 Report

Despite the authors' partial response to my suggestions, I believe the paper can still be considered for publication. They may receive slightly fewer citations as a result, but that is acceptable.